# Chitinase-Assisted Bioconversion of Chitinous Waste for Development of Value-Added Chito-Oligosaccharides Products

**DOI:** 10.3390/biology12010087

**Published:** 2023-01-05

**Authors:** Siriporn Taokaew, Worawut Kriangkrai

**Affiliations:** 1Department of Materials Science and Bioengineering, School of Engineering, Nagaoka University of Technology, Nagaoka, Niigata 940-2188, Japan; 2Department of Pharmaceutical Technology, Faculty of Pharmaceutical Sciences, Naresuan University, Phitsanulok 65000, Thailand

**Keywords:** chitinase, chitin, waste, chitosan, chito-oligosaccharides, bioconversion, sustainability

## Abstract

**Simple Summary:**

Bioconversion of chitinous waste to chito-oligosaccharides using chitinase is an attractive strategy for traditional waste management. Chito-oligosaccharides have a broad range of applications due to their water solubility and possess various biological properties. The different sources of chitinase provide different yields and physicochemical properties, e.g., the degree of polymerization of chito-oligosaccharides. This review discusses the potential of chitinase in chito-oligosaccharide production with a focus on the chitinase sources, chemo-enzymatic production of chito-oligosaccharides and their derivatives, applications of chito-oligosaccharides, and the route to industrialization, based on the academic studies published within the most recent decade.

**Abstract:**

Chito-oligosaccharides (COSs) are the partially hydrolyzed products of chitin, which is abundant in the shells of crustaceans, the cuticles of insects, and the cell walls of fungi. These oligosaccharides have received immense interest in the last few decades due to their highly promising bioactivities, such as their anti-microbial, anti-tumor, and anti-inflammatory properties. Regarding environmental concerns, COSs are obtained by enzymatic hydrolysis by chitinase under milder conditions compared to the typical chemical degradation. This review provides updated information about research on new chitinase derived from various sources, including bacteria, fungi, plants, and animals, employed for the efficient production of COSs. The route to industrialization of these chitinases and COS products is also described.

## 1. Introduction

Globally, about six to eight million tons of sea food waste is generated from crab, shrimp, and lobster shells every year [1]. The environmental concerns are caused by the disposal of this seafood waste accumulating along coastal areas since their natural digestibility is relatively low. In developed countries, disposal is costly—up to USD 150 per ton in Australia, for example [2]. Because such waste is a rich source of chitin (15–40% of the total weight of crustaceans), not only is the bioconversion of chitin food wastes to added-value products, an effective, eco-friendly form of waste management, but also those products are valuable from a market perspective [3,4]. For instance, dried shrimp shells are valued at USD 100–120 per ton. The ground shells can be used as an animal-feed supplement, as fertilizer, and in chitin production [2].

Chitin is the second most abundant polysaccharide. It is composed of repeated units of N-acetylglucosamine (GlcNAc) linked by β-(1,4)-glycosidic bonds for 100% degree of acetylation (DA), or the DA is lower due to the composition of D-glucosamine (GlcN) [5]. Predominantly, chitin is found in the cell walls of fungi [6,7,8,9,10,11,12] and algae [13], the exoskeletons of invertebrates, including insects [14,15,16], and the structural component of crustaceans [4]. Due to inter- and intra-molecular hydrogen bonding, chitin is insoluble in water and common organic solvents but requires solubilizing agents [17,18,19]. Depending on the internal chain arrangements, chitin is classified into α-, β-, or γ-forms with different packing patterns and physicochemical properties. Structures of α and β chitins are anti-parallel and parallel chain arrangements, respectively, whereas γ-chitin is structurally similar to α-chitin, having hydrogen bonds to stabilize the crystal unit in two directions [20,21]. The α-chitin is the most tightly packed and is insoluble in water, while β- and γ-chitins are more water accessible, with more flexible structures [22]. However, there are debates whether the γ form is a new form or is an analogue of the α form but with mixed packing [23]. The chitin structure in native environments and cell walls of algae and fungi, for example, is much more complicated than the simplified models, which are determined using high-crystallinity and purified chitin materials. This is due to the complexity of chitin synthesis and the complex environments of chitin and other cell-wall components [12,24]. In seafood waste such as shrimp- and crab-shells, chitin is mainly found in the form of α-chitin [22].

The hydrolysis of chitin yields chitosan, which is of immense significance due to its vast biological applications. Chitosan with a degree of polymerization (DP) between 2 and 10 and an average molecular weight (MW) less than 3.9 kDa is called chito-oligosaccharides (COSs), chitosan oligomers, or chito-oligomers [25]. The COSs generally consist of GlcNAc or GlcN units. They can be produced with defined DP and different DA (generally lower than 25%); however, their pattern of acetylation (PA) is always random but can be specified by using a combination of chitin oligosaccharide deacetylases [26]. Homo-chito-oligosaccharides are the oligomers of GlcN or GlcNAc (fully acetylated or fully deacetylated COS), while hetero-chito-oligosaccharides, comprise a combination of GlcN and GlcNAc units (partially acetylated COS), with variations in the DP, DA, degree of deacetylation (DD), and position of N-acetyl residues in the oligomer chain [27]. The size of COS defined by DP, DD (or DA), and PA influences their properties [28]. Due to many active groups, such as –NH_2_, –OH, and a small number from the N-ethylphthalide amino group, COS can dissociate in aqueous solutions and form intra- and intermolecular hydrogen bonds [29]. Hetero-chito-oligosaccharides with DP less than 10 are typically water soluble; however, water solubility of COS with a DP of more than 10 depends on the DA and the pH of solution [30,31]. The water-soluble COSs can be used in pharmaceutical and biotechnological applications [32], but temperature- and pH-tolerances of COSs are important in maintaining their biological activity [33].

The COSs are produced by chemical or enzymic processes. The chemical processes for demineralization, deproteinization, and discoloration involve the use of environmentally hazardous chemicals, e.g., hydrochloric acid and sodium hydroxide at a high concentration [34]. Due to the toxicity caused by the chemical process, COS products are not suitable for human consumption. In contrast, enzymatic approaches carried out under mild conditions—e.g., at ambient temperature and neutral pH—by chitinase, which hydrolyzes the β-(1,4)-glycosidic bonds in a chitin chain, have gained more attention [35]. This strategy provides COSs as products that are environmentally friendly and safe for human usages, enabling them to be applied in various fields of agriculture [36,37,38], biotechnology [39,40,41], and biomedicine [42,43,44].

Therefore, exploiting enzyme-based technology to produce bioactive COSs of much greater value for commercial use is a promising approach for the disposal and recycling of enormous chitin-containing waste. The recent research literature on new chitinase from various sources of COS production, COS-derivative synthesis, updated applications, and commercialization of COSs are summarized in this review.

## 2. Sources of Chitinase and COS Production Efficiency

### 2.1. Bacteria

In recent years, enzymatic degradation of chitin has been of increasing interest because of its green characteristics. The degradation by chitinases is the key step in chitin degradation. Extracellular chitinase-producing bacteria use chitin or colloidal chitin as a carbon source for the production of a mixture of chitinases and N-acetylglucosaminidase [45]. The newly formed chitinase facilitates the production of COSs. Bacterial sources of chitinase for the production of COSs from different chitin substrates are summarized in Table 1. Numerous bacterial chitinases have also been cloned and expressed in *Escherichia coli*, *Pichia pastoris*, *Manduca sexta*, and *Bacillus subtilis* [46,47].

Wang and Li et al. [48] studied chitinase *BsCh*i from *Bacillus subtilis*, which was recombinantly expressed in *Escherichia coli* and characterized as a potent chitinase for degrading crystalline chitin substrates such as α-chitin, β-chitin, and crab shells. The *BsChi* has lower activity toward α-chitin, similar to other chitinases such as chitinase from *Chitiniphilus shinanonensis* [49], *Enterobacter cloacae* subsp. *Cloacae* [51], and *Paenibacillus* sp. *LS1* [54]. The decrease of *BsChi* activity toward α-chitin in reaction time by 22-fold (from 7.5 mM/min at the first hour) is more obvious than that toward β-chitin, which decreases by 3-fold. The productivity of *BsChi* to (GlcNAc)_2_ at 12 h can be increased over 200% by both mechanical and protease pretreatment of the crab shell [48]. This suggests a strict substrate specificity. On the other hand, chitinase genes *PbChi67*, *PbChi70*, and *PbChi74* produced from *Paenicibacillus barengoltzii*, a marine bacterium, have different specificities. The genes *PbChi67* show the specific activity towards chitosan [56]. Meanwhile, the genes *PbChi67* show high activity towards chitin, similar to *PbChi70,* which has enhanced specific activity towards colloidal chitin (30.1 U/mg) that is 60 times higher than the activity toward crab shell powder. Furthermore, it is highly active towards (GlcNAc)_5_ (213.4 U/mg) but displays no activity towards (GlcNAc)_2_ [55]. The substrate specificity of chitinase for shrimp shell chitin is higher than that of crab shell chitin. For chitinase from *Cellulosimicrobium funkei* HY-13, the specific activities of *rChiJ* for colloidal shrimp shell chitin and colloidal crab shell chitin are 16.0 U/mg and 8.1 U/mg, respectively [65]. However, (GlcNAc)_2_ is obtained as the major end product regardless of the used substrates, i.e., shells of shrimp, chitin, and cray fish [52,65]. For the long-chain COSs, transglycosylation (TG) of chitinases has gained much attention for use in agricultural applications. Chitinases from *Enterobacter cloacae subsp*. *Cloacae* (*EcChi1*) [51], *Flavobacterium johnsoniae* (*FjChiB*) [53], and *Serratia marcescens* (*SmChiD*) [66] have TG activity, which can transfer the released COSs to a suitable acceptor, rather than using a water molecule to break the glycosidic bond. Therefore, the products are longer than donor oligosaccharides. *EcChi1* and *SmChiD* exhibit high TG activity on (GlcNAc)_2–6_ as substrates. With (GlcNAc)_3_ as a starting substrate, TG products (GlcNAc)_4–7_ are detectable, whereas (GlcNAc)_9_ is detectable as the longest TG product from (GlcNAc)_5–6_ as substrates [51]. *FjChiB* exhibits transient TG activity on (GlcNAc)_5_ and (GlcNAc)_6_ substrates for (GlcNAc)_6–8_ and (GlcNAc)_7–9_ products, respectively [53].

Recently, chitinases from *Streptomyces* species have been reported and biochemically characterized such as *Streptomyces albolongus* [57], *Streptomyces chilikensis* [58,59], *Streptomyces diastaticus* [60], *Streptomyces luridiscabiei* [67], and *Streptomyces sampsonii* [61]. The genes encoding chitosanases from *Streptomyces albolongus*, *Streptomyces diastaticus,* and *Streptomyces sampsonii* are cloned, sequenced, and expressed in *Escherichia coli*. Chitinases from *Streptomyces albolongus* and *Streptomyces sampsonii* degrade chitin to (GlcNAc)_2–3_ at the initial period of hydrolysis and primarily GlcNAc and (GlcNAc)_2_ as final products, similar to chitinase from other sources such as *Salinivibrio* sp. [62], *Thermomyces lanuginosus* [63], and *Vibrio campbellii* [64]. However, (GlcNAc)_2–5_ products are obtained by *Streptomyces diastaticus* [60]. When compared to similar chitin starting materials (1 to 2 g), chitinase from *Vibrio campbellii* generates a higher yield of (GlcNAc)_2_, being 2-fold greater than the yield produced by chitinase from *Thermomyces lanuginosus* [63], 10-fold greater than that produced by chitinase from *Paenicibacillus barengoltzii* [55], and 100-fold greater than that produced by chitinase from *Streptomyces sampsonii* XY 2–7 [61].

### 2.2. Fungi

Varieties of chitinases can be found among fungal species, and each species can produce different chitinase isomers, which have different catalytic properties. Chitinases are not only involved in exogenous chitin decomposition but also in fungal cell-wall degradation and morphogenesis, in which the cleave of chitin is crucial for hyphal growth, septum formation, and spore germination [68,69]. Fungal chitinases can be used for the treatment of chitinous waste from fisheries. For instance, the filamentous fungus *Aspergillus niveus* produces extracellular antifungal chitinase to degrade crab shells by using the shells as the carbon source [70]. *Aspergillus* sp., such as *Aspergillus flavus* isolated from soil, has a potential for the efficient production of GlcNAc as the main product (~80% GlcNAc and ~20% of (GlcNAc)_4_), similar to mesophilic *Penicillium monoverticillium* CFR 2 and *Fusarium oxysporum* CFR 8. In 48 h, when 10 mg/mL colloidal chitin and crystalline α-chitin are used, the concentrated crude chitinase from these fungi yield about 90 and 10 mmol/L of GlcNAc, respectively [71]. However, in 30 min, GlcNAc concentrations are about 20 and 2–3 mmol/L, respectively. This prolonged incubation results in the degradation of (GlcNAc)_2_ and (GlcNAc)_3_, similar to the result of incubating with crude chitinase from *Humicola grisea* [72]. As compared to *Fusarium* sp., *Paecilomyces lilacinus* (EF183511) expresses a higher chitin digestion ability with a chitinase activity of 60 U/mL [73]. This means that the major product might be GlcNAc. Krolicka and Hinz et al. [74] reported production and characterization of Chitinase from *Myceliophthora thermophila* C1 fungus for bioconversion of chitin waste to COS. They found that this chitinase exhibits high thermostability at 40 °C (>140 h 90% activity), 50 °C (>168 h 90% activity), and 55 °C (half-life 48 h), whereas degradation rates of COSs are slow. The calculated rates for the degradation of COS are 0.02, 0.2, 0.17, and 0.18 mM/min for (GlcNAc)_3–6_, respectively [74]. Similar results are found using chitinase from *Rhizomucor miehei* and *Trichoderma harzianum*, which can hydrolyze colloidal chitin to fully acetylated COS [75,76].

### 2.3. Plants

Plants have been known to synthesize various types of chitinases, which have various physiological functions including self-defense, growth, and stress tolerance. These defensive enzymes hydrolyze β-1,4-glycosidic linkages of chitin localized in the cell wall of pathogens such as fungi [77]. In other words, the plant chitinases exhibit antifungal activity. In the plant kingdom, chitinase consists of two N-terminal lysin motif (LysM) domains and a C-terminal catalytic (cat) domain of the glycoside hydrolase family 18 (GH18). The LysM domains are found in receptors which recognize chitin elicitors from the cell walls of fungal pathogens. Loss of the LysM domain or the hydrolytic activity is shown to abrogate the antifungal activity of the chitinases [78]. The LysM-domain multimer fusion chitinases (LysMn-Cat, n = 1–4) hydrolyze chitin more efficiently than the single catalytic domain and exhibit a stronger antifungal activity than LysMn (Figure 1A) by attacking the tips and lateral walls around the septa of the fungal hyphae (Figure 1B–E). The mode of interaction of LysM in the chitinases among plants may differ, for example, LysM acting in chitin-signaling immunity in fern and higher plants, which have been extensively studied. For subtropical plant species, i.e., fern and moss, the recent research reported LysM-containing chitinases from *Pteris ryukyuensis* [78,79], *Equisetum arvense* [80,81], *Selaginella doederleinii* [82], and moss *Bryum coronatum* [83]. The LysM domains of chitinase *PrChiA* from *Pteris ryukyuensis* are demonstrated to bind to chitin and COS. (GlcNAc)_6,_ (GlcNAc)_5_, and (GlcNAc)_4_ are the most frequently hydrolyzed into (GlcNAc)_4_+(GlcNAc)_2,_ (GlcNAc)_3_+(GlcNAc)_2_, and (GlcNAc)_3_+GlcNAc, respectively. The enzyme hardly catalyzes the TG reaction [79]. Under the same conditions, the chitin-binding ability of *PrChiA* is lower than chitinase from *Equisetum arvense* (*EaChiA*), but the binding mode of these two chitinases for (GlcNAc)_n_ are comparable [80,81]. The cleavage patterns of (GlcNAc)_n_ produced by chitinase from *Selaginella doederleinii* (*SdChiA*) are also similar, except for (GlcNAc)_4_ which is mainly hydrolyzed to (GlcNAc)_2_ [82].

### 2.4. Animals

Although mammals do not produce chitin, mice and humans synthesize two active chitinases to digest chitin in animal bodies. Those two chitinases include chitotriosidase (*ChiT1*) and acidic chitinase (*ChiA* or *AMCase*), which belong to the GH18. The acidic chitinases which exhibit chitinolytic activity and degrade chitosan to produce COS under stomach conditions are found in porcine [84,85], mouse [86,87], chicken (*Gallus gallus domesticus*) [88], marmoset (*Callithrix jacchus*) [89], and crab-eating monkey (*Macaca fascicularis*) [90,91]. *AMCase* is predominantly found in pig stomach tissue. Due to *AMCase* functioning as a protease-resistant glycoside hydrolase in the pig digestive system, the content of *AMCase* is much higher than that of *ChiT1*. In healthy pig eyes, mRNA expression of *ChiT1* is ten times higher than that of *AMCase*. This implies that *ChiT1* protects mammalian eyes from chitin-containing pathogens. In certain ocular pathologies, *AMCase* may be a mediator of innate immune responses [84].

Expression and activity levels of acidic chitinases are much higher in omnivorous animals in comparison with carnivorous and herbivorous animals. In particular, *Macaca fascicularis ChiA* exhibits higher chitinolytic activity than mouse *ChiA* at broad pH (1.0–7.0) and temperature (30–70 °C) ranges [91]. In the case of pH stability, because of the pH shift from 2.0 to 5.0–7.0 after feeding, the chitinolytic activity might relate to its principal expression and localization in the stomach. Activity variability between chitin and chitosan substrates is driven by the differential substrate specificity of *ChiA*. Thus, the composition of resultant degraded products can be altered. Chitin and (GlcNAc)_5_ are degraded into mainly (GlcNAc)_2_ and GlcNAc monomers, respectively, by porcine *ChiA*, whereas the GlcN substrates remain constant. It indicates that, unlike chitosanases, GlcN–GlcN bonds are not hydrolyzed by the porcine *ChiA*. After the treatment of chitosan with porcine *ChiA*, the main products are similar to (GlcNAc)_3_ or hetero-chitotrimers such as GlcN-GlcNAc-GlcNAc and GlcNAc-GlcN-GlcNAc [85]. In a report by Uehara, et al. [90], monkey *ChiA* produced the highest levels of (GlcNAc)_2_ from chitin, followed by random-type chitosan, α-chitin, colloidal chitin, and block-type chitosan (Figure 2A,B). For chitosan substrates, the pattern of the produced COSs is different between the chitosan types (Figure 2B). The size distribution of the produced COSs is shown in Figure 2C. *ChiA* preferentially degrades the randomly placed GlcNAc-rich regions to produce more variably sized COSs. Simultaneously, it targets a cluster of the acetylated areas (GlcN-rich) of the block-type chitosan and produces mainly dimers and trimers [86].

## 3. Chemo-Enzymic Production of COSs and Its Derivatives

### 3.1. Chemo-Enzymic Production of COSs

Since chitin is hydrolyzed by chitinase at the solid–liquid interface, the pretreatment of chitin is required to enhance enzyme accessibility to control the hydrolysis rate of chitin and COS product yield [92]. The conventional method includes the use of concentrated hydrochloric acid, sulphonic acid, or sodium hydroxide to obtain colloidal chitin [93]. Due to environmental concerns, effective, green pretreatment methods such as a pretreatment using ionic liquids have gained attention. Ionic liquids (ILs) are molten salts that consist of bulky organic cations and inorganic/organic anions, which can dissolve carbohydrate polymers. They are well known as emerging green solvents due to their low volatility and recyclability, which minimizes the solvent consumption in the environment [94]. Hence, recent research investigates chemical pretreatment of chitin using ionic liquids followed by enzymatic hydrolysis. Li and Huang et al. [92] reported that chitin pretreated by 1-ethyl-3-methylimidazolium acetate IL provided an efficient production of GlcNAc (175.62 mg/g chitin) and (GlcNAc)_2_ (341.70 mg/g chitin) with a conversion of 61% at 48 h after catalysis by chitinase from *Streptomyces albolongus*. However, colloidal chitin pretreated by hydrochloric acid allows the highest hydrolytic activity and conversion rate at 88%. By using 1-Ethyl-3-methylimidazolium bromide (EMB) and Trihexyltetradecylphosphonium bis(2,4,4-trimethylpentyl)phosphinate (TBP) ILs to pretreat chitin and then directly hydrolyze the chitin using chitinase from *Thermomyces lanuginosus* ITCC 8895, 78 and 107 mg (GlcNAc)_3_/g chitin, respectively, are achieved in the shorter time (2 h) [95].

### 3.2. Production of COS Derivatives

The structures and properties of chitosan and its derivatives have been extensively studied to improve their structural properties for a particular application. However, research on the synthesis of COS derivatives and identification of their enhanced biological activities has also gained interest recently. Ngo et al. [96] synthesized aminoethyl-chito-oligosaccharides (AE-COS) by chemical reaction of COS (MW 0.8–3 kDa, DD 90%) using 2-chlorethylamino hydrochloride at 40 °C for grafting the aminoethyl group at C-6 onto COS. In the structure of AE-COS, the original frame of COS is maintained because the C-6 hydroxyl groups have the highest reactivity for aminoethylation [97]. The AE-COS exhibits an inhibitory effect on myeloperoxidase activity about 20% higher than COS at 100 μg/mL and an apoptotic effect on the growth of cancer cells such as human gastric adenocarcinoma (AGS) [98], human fibrosarcoma [97], and murine microglial BV-2 [99] cells. The action mechanism can be caused by the functional group of aminoethyl with a positive charge attached to AE-COS, reducing the expression of matrix metalloproteinases (MMPs) [97], nitric oxide synthase (iNOS), and cyclooxygenase-2 (COX-2) [99]. Chemical modification of COS has become a method to overcome the limitation of other sensitive materials such as proteins. For instance, poor stability, low cell affinity, and instable activity of endostatin2 (ES2) used as the antitumor drug are improved by conjugation with COS derivatives to produce O-(2-hydroxyl) propyl-3-trimethyl ammonium chito-oligosaccharide chloride (HTCOSC) [100]. This COS derivative is prepared through the reaction with glycidyl trimethylammonium chloride, and then conjugated with ES2. However, one molecular HTCOSC can be conjugated with one molecular ES2 on average in a HTCOSC-ES2 polymer. This is because the large molecular size of HTCOSC (MW 3 kDa) causes steric hindrance after one molecule of HTCOSC is introduced to ES2 (MW 1223 Da). Thus, binding of a second molecule of HTCOSC to ES2 becomes much more difficult. Accordingly, when the C-terminal carboxyl group is exposed, the reaction is carried out mainly between C-terminal carboxyl and C-2-NH_2_ of HTCOSC [101]. The half-life of HTCOSC-ES2 is prolonged, and its bioavailability is improved after single intravenous administration to mice, compared to those of ES2. In a tissue distribution study, both ES2 and HTCOSC-ES2 reach a maximum concentration at 2 h in liver but lower concentrations in other tissues. This indicates that HTCOSC-ES2 exhibits liver targeting. However, the HTCOSC-ES2 concentration in the liver is higher than that of ES2 [100]. It is noted that there is no report about the metabolism, immunogenicity, and toxicity of HTCOSC-ES2 when it is used alone or in combination with other agents [101].

The phenolic acids are known to donate an H-atom leading to enhanced antioxidant and anti-inflammatory properties of COSs. For medicinal purposes, COSs grafted with phenolic compounds such as gallic acid and epigallocatechin-3-gallate have been studied. Antioxidant activity of epigallocatechin-3-gallate (EGCG) is relevant to the abilities of free radical quenching, a chain reaction terminating in lipid oxidation, and metal-ion chelating. Phenolic acid-conjugated COS can be synthesized by a free-radical grafting method. Initially, hydroxyl radicals are generated by a redox pair reaction between ascorbic acid and H_2_O_2_. These radicals oxidize amino or hydroxyl groups of COS glucosamine units, resulting in the formation of macroradicals. The EGCG is then formed by covalent bonding between the radicals localized on COS and phenolic acids [102].

The conjugation between COS and EGCG occurs at C-2, 3 or 6 of COS and can increase the number of hydroxyl groups leading to enhanced antioxidant activities of COS [102]. The conjugation of gallic acid and gallate using COS having an MW range of 3–5 kDa or a DP of 3–5 has been synthesized for the prevention of cancers. Vo and Ngo et al. [103,104] reported that leukemia and lung cancer (*in vitro* tests with rat basophilic leukemia (RBL-2H3) and human lung epithelial A549 cells, respectively) were inhibited by gallic acid-grafted-COS. Under the same preparation method, the gallic acid-grafted-COS negatively influenced the proliferation of human gastric cancer cells (adenocarcinoma AGS cells) [105].

## 4. Applications of COS

### 4.1. Food Additives and Functional Food

Treatment with COS has been a topic of several preclinical studies suggesting that it is safe for humans to consume. In functional food, COS increases the stability of curcumin-loaded pickering emulsions through electrostatic interactions between COS and glycated whey protein isolate to form nanoparticles [106]. As bioactive molecules, various anti-tumor and anti-inflammatory activities of COSs have been reported. They are also involved in the treatment of certain metabolic disorders of which the conditions include increased blood pressure, high blood sugar, excess body fat around the waist, and abnormal cholesterol or triglyceride levels. Potential antihyperlipidemic activity of COSs as a functional food/dietary supplement has been shown to lower cholesterol level in rats fed a high-fat diet [107,108,109,110] and in healthy men (smokers and non-smokers) [111]. In an *in vitro* study, two types of COSs with different molecular weights (1 kDa and 3 kDa) reduce intracellular lipid accumulation, decrease triglyceride level in human hepatocellular carcinoma (HepG2) cells [112], and inhibit adipogenesis in 3T3-L1 adipocytes [113]. The COSs may be effective as preventive agents in fatty liver disease.

COSs (MW < 6 kDa, DD 88%) inhibit non-pathogenic human colonic bacteria weakly, while the growth rate of *Bifidobacteria* of human origin is not affected [114,115]. Applications of COSs as dietary supplements, prebiotics, and biopreservatives in dairy products and beverages have been proposed. For the growth of lactic acid bacteria, COSs (DP 4–9, MW < 1.7 kDa, DD 60%) at 0.1% *w/v* stimulate the growth of *Lactobacillus paracasei* and *Lactobacillus kefir*, but the higher MW COSs inhibit their growth [116]. For yoghurt prepared by a culture of *Streptococcus thermophilus* and *Lactobacillus delbrueckii* subsp. *Bulgaricus*, COS (MW 2.7 kDa, DD 68%) can be incorporated into the fermentation at a maximum concentration of 0.1% *w*/*w* and provided good sensory acceptance of the yoghurts. Higher concentrations cause metabolic alterations in lactic acid bacteria, and the yoghurt product does not fulfill the standards defined for yoghurt by the Codex Alimentarius [117]. The ability of COSs (2 or 3 kDa, DD 70%) to prevent the formation of staling compounds, including 5-hydroxymethylfurfural, trans-2-nonenal, and phenylacetaldehyde, in beer is reported at 0.001–0.01%. Furthermore, COSs reduce contamination of beer-spoilage microbes, e.g., *Lactobacillus brevis,* in the brewing process and show radical scavenging activity in the finished beer. This improves the shelf-life stability by allowing DPPH/hydroxyl radical scavenging during beer storage [118,119].

### 4.2. Biomaterials and Biomedicines

The COSs are promising as drug candidates, since they are naturally biocompatible, non-toxic, and non-allergenic to living tissues. Their anti-tumor properties, such as the inhibition of liver tumor cell metastasis, have also been investigated [120]. In *in vitro* studies, COS increases apoptosis in human acute leukemia HL-60 cells [121], and human renal carcinoma [122]. In *in vivo* studies, oral or intraperitoneal administration of COSs has shown anti-cancer activities in a C57BL/6 mouse model of colitis-associated colorectal cancer [123] and lung cancer [124]. The COS decreases tumor volume from about 1000 to 200 mm^3^ when 100 mg COS/kg is used with tumor-bearing mice [124]. Johansen and Carretta et al. [125] found that COS exerted their anti-tumorigenic effects through the inhibition of YKL-40 (also known as chitinase-3-like 1 or *Chi3L1*), which is a conserved, secreted chitinase-like protein (CLP) found in elevated levels in many diseases, including cancer [126,127].

The most common biomedical applications of COS are their use as adjuvants, in particle or gel form, or as drug delivery systems for metabolic or anti-neoplastic agents [128]. It is also used to conjugate with other materials to improve surface biocompatibility such as fibroin [129] and poly(ester-urethane) [29]. To meet an important requirement for *in vivo* biomedical applications, the erosion rate being controlled by DP of COSs is studied. Ailincai and Rosca et al. [130] prepared porous hydrogels from COS with different DPs (14 to 51), by crosslinking with 2-formylphenylboronic acid in three molar ratios of their functionalities and studied the erosion rate. The hydrogels possess *in vitro* enzymatic degradability, due to COS presence. The biodegradation rate depends on the DP of the shortest COS; the mass loss increases from 16 to 43%. *In vivo* and *ex-vivo* biocompatibility investigation on mice shows no cytotoxic effect, and *in vitro* antimicrobial tests reveal remarkable antimicrobial properties on a Gram-positive bacterial strain, *Staphylococcus aureus*; a Gram-negative bacterial strain, *Escherichia coli*; four yeast strains: *Candida albicans*, *Candida parapsilosis*, *Candida glabrata*, and *Saccharomyces cerevisiae*; and three fungal strains: *Penicillium chrysogenum*, *Cladosporium cladosporioides*, and *Aspergillus brasiliensis* [130]. Although the antimicrobial properties of COSs are well-known, beneficial effects on probiotic bacteria such as *Bifidobacterium* species and other gut microbiota are also reported [131,132]. This suggests the prevention of colitis-associated colorectal cancer development [123]. Furthermore, COSs induce apoptosis of colorectal cancer SW480 cell line and retain a radiation-sensitive killing effect. This is beneficial for the therapeutic effect via radio therapy for colon cancer [133]. Similarly, radiosensitization effects of COSs on human lung cancer line HepG2 are also reported [134]. For the protective effect, vision-threatening diseases are prevented by COS. Fang and Yang [135] found that COS prevented retinal ischemia and reperfusion injury through oxidative stress and inflammation inhibitions.

### 4.3. Plant Elicitors

Much attention has been garnered by COSs due to elicit defense responses in various plants. In tobacco, the defense response is induced by nitric oxide pathways [136]. In rice, the immune responses to microbial infection are induced by LysM modules, which perceive peptidoglycan fragments from bacterial pathogens or chitin fragments from fungal pathogens, and then elicitor signals are transduced [137,138]. In a comparison of COSs with DP between 5–7, the DP 7 strongly induces phenylalanine ammonia lyase and Isochorismate synthase -1 genes, with a concomitant increase of mitogen-activated protein kinase 6 and WRKY45 transcription factor genes. This indicates a defense response in rice seedlings [139]. In tomatoes, COS has synthetic fungicides in the control of the devastating fungal pathogen *Botrytis cinerea*, the causal agent of grey mold disease. The combined treatment with COS (200 mg/L ε-poly-l-lysine+ 400 mg/L COS) provides optimal *in vitro* antifungal activities, achieving an inhibition rate of 90.22%. *In vivo* assays with these combined bio-fungicides under greenhouse conditions demonstrate good protection against severe grey mold [140].

When COS has an MW of ~3.5 kDa and DD of ~70%, it enhances barley germination for improving the quality of malt. The malt quality is improved by COS in seed priming at 1 mg/L [141]. The different DP of COSs affects the growth and photosynthesis parameters of wheat seedlings. The COSs with an MW < 3 kDa and DP > 3 promotes the growth and photosynthesis of the wheat [142]. This defense response of wheat seedlings is also shown under salt stress [143]. However, COSs with an MW of 2 kDa and DP 8–16 increase the content of chlorophyll of coffee [144]. Cross-protection roles of COS in alleviating the toxicity of cadmium (Cd), which is contained in a wide range of applications such as pesticides, herbicides, fertilizers, and industrial wastewater in edible rape (*Brassica rapa* L.), are studied. The COS with MW of 1.6 kDa and DD 82%) at 50 mg/L relatively reduces the Cd concentration in shoots and roots of the rape under a Cd stressed condition (Figure 3) [145].

## 5. Route to Industrialization of COSs by Bioconversion from Chitinous Waste

The enzymatic production of COS products from seafood waste represents an attractive alternative to traditional waste management. In particular, an awareness of the known advantages of using chitinase in the production of COSs is growing exponentially. Commercial interest in chitinase has continued to increase, demonstrated by the increased number of patents filed worldwide since 2000 (Figure 4). Academic publications indicate that enzymatic productions of COSs by an immobilization system to improve stability and reusability of the enzyme have also generated recent interest [146,147]. On the other hand, chitinase-producing bacteria such as *Pseudomonas gessardii* are directly immobilized and used for degrading chitin. This method is patented and commercialized for a plant protection product, such as a fungicide, biostimulant, or insecticide, by the BioMosae company, the Netherlands [148]. Similarly, for agricultural purposes, TCI Products company, USA, patented a fermented liquid mixture of crab and shrimp shell powder with chitinase produced by *Bacillus pumilis* and other bacteria that produce other enzymes for a biological pesticide [149]. As a fertilizer, soil containing fermented crab and shrimp shells with chitinase-producing bacteria, i.e., *Bacillus licheniformis* and *Paenibacillus ehimensis,* is commercialized by Osprey Biotechnics Inc., USA. Due to the composition of nitrogen and phosphorous in crustacean shells, it is readily available to a growing plant, resulting in improved growth rates, and/or higher crop yields [150]. However, in recent years, the patents related to COS production from chitin-containing waste by using chitinase are mainly filed by academic institutions in China. One example of a patented method is COS production from shrimp shells by using chitinase produced by *Exiguobacterium antarcticum* DW2 and *Paenibacillus chitinolyticus* [151,152,153]. A method for preparing COS by utilizing other sources of chitinous waste such as mycelia residues from citric acid fermentation is also patented. The invention relates to a method for extracting chitin from the fungal waste in citric-acid fermentation, and then degrading chitin with chitinase [154].

Since the pilot/large-scale enzymatic productions of COS using chitinous waste have rarely been reported, the feasibility of production on a large scale has been assessed based on manufacturing costs by techno-economic analysis (TEA) from a simulation perspective [155]. The TEA of bioconversion of COSs from chitinous waste using chitinase has not been fully developed, possibly due to the challenge of production costs in enzyme production. However, the pretreatment of seafood waste to extract chitin and chitosan, which is the initial step in COS production, can be used as a guideline. Gómez and Barrera et al. [156] reported the TEA of chitosan production from shrimp shell waste by using Aspen Plus^®^ (Figure 5). The processes to obtain chitin include size reduction (CR-100), depigmentation (T-100 and F-100), deproteinization (R-100 and F-200), demineralization (R-200), and filtration of chitin (F-300). The prices of the inlet stream are based on the reactant costs, which are an ethanol-water solution (0.85 USD/kg), sodium hydroxide (0.2 USD/kg), and hydrochloric acid (0.3 USD/kg) in depigmentation, deproteinization–deacetylation, and demineralization, respectively [157]. In shrimp biorefinery, the major contributor is the cost of raw materials at USD 24.71 MM per year. Due to the zero-cost raw material (shrimp shell waste), this cost is deducted from the cost of the fresh shrimp (5.7 USD/kg) with a mass flow rate of about 4110 ton/year [158]. Since this conventional chemical chitin production method addresses the use of corrosive reagents and the generation of a large amount of waste, the biological pretreatment of shrimp shells or the use of a hot water-carbonic system are also a topic of study for chitin production. Yang and Gözaydın et al. [159] conducted life cycle assessment of this conventional pretreatment process compared with a hot water-carbonic system. They reported that this conventional process generates corrosive acidic or alkali wastewater, which requires around 60 kWh for wastewater treatment and contributes to the carbon footprint by 283 kg CO_2_ equivalent for every 100 kg of shrimp shell. Some research has focused on mass integration to conserve resources by considering design strategies for energy integration, material conservation, and wastewater reduction. In a shrimp-based biorefinery, the freshwater demand may exceed 300 Million tons/year if the foreseen global market of shrimp reaches 7.28 Million tons by 2025 [160], which also means the generation of a large amount of waste. Moreno and Martínez et al. [160] showed that the integrated biorefinery has the capacity to process 469.22 kg/h of fresh shrimp while reducing fresh resource consumption by 56.6% with a 53% waste reduction compared to the non-integrated approach. The chitin and chitosan production rates are increased up to 3.8 and 3.1 kg/h, respectively, by the material recovery in residual water recycling. The environmental study shows reductions in atmospheric and toxicological categories estimated at 7.8 and 3.5%, respectively, for potential environmental impacts (PEI).

## 6. Conclusions and Outlook

A promising strategy for recycling an enormous amount of chitinous waste is the bioconversion to added-value products, i.e., COSs by using chitinase. Recent literature studies the novel sources of chitinase produced by bacteria, fungi, plants, and animals, as well as their activities in COS production. In the pretreatment of chitinous waste, ionic liquid technology has gained attention in replacing the typical concentrated strong acids and bases. Due to their high water solubility and active biological properties, including anti-microbial, anti-cancer, and anti-oxidant properties, COSs can be applied in a broad range of applications. New applications of COSs are being studied, especially as functional food/additives and biomedicines in cancer therapies, whereas studies of their application as plant elicitors/fertilizers are continuing. While the academic publications of COSs using chitinase are growing exponentially, its techno-economic analysis, the promising development directions towards industrialization, and pilot-scale studies are rarely found.

Even though novel sources of chitinase, especially from bacteria, are currently being studied, only a few genome-sequencing data sets with chitin-degradation machinery are reported in the chitinolytic bacteria capable of degrading the chitin. Increasing collective knowledge through whole-genome sequencing investigations of diverse chitinolytic strains will aid in revealing the genomic basis of chitin-degradation activity and the application of degraded products of chitin such as COSs.

For an efficient production of COSs, pretreatment of chitinous waste or chitin by chemical degradation, followed by enzymic hydrolysis, is required. Alternatively, pretreatment using ionic liquids, more specifically, a green solvent, is proposed. However, an obstacle is the high cost of ionic liquids. Hence, studies of recovery and purification of ionic liquids in the pretreatment of chitin for COS production should be included.

Besides the production of high-molecular weight COS through transglycosylation reactions, it is essential to develop a novel and reproducible method of producing COSs with specific DP, DA/DD, MW, and PA to advance the knowledge of biological activities. In order to understand the relationships between physicochemical and biological properties of COSs, characterization is completely prerequisite. Generally, COS synthesized by the enzymatic method are obtained as a mixture of oligomers. The complete structural characterization and separation of COSs from the reaction mixture are a challenging task. However, well-characterized, pure, concentrated fractions of COSs are required to evaluate their biological activities and to use them in synthesis of rare sugars, for example, heterogenous sugars such as (GlcNAc-GlcN) or (GlcN-GlcNAc), various COS derivatives, and other functionalized COS-based nanomaterials. Importantly, high quality, purity, and accurate quantification of COS are demanded in biomedical and food applications.

## Figures and Tables

**Figure 1 biology-12-00087-f001:**
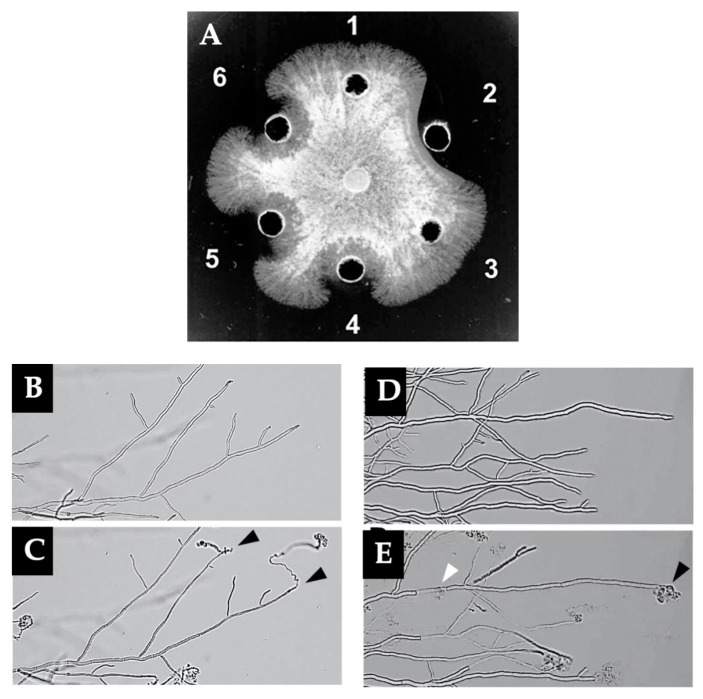
Antifungal activity against *Trichoderma viride* of LysMn and LysMn-Cat. (**A**) Well 1, control (distilled water); Well 2, LysM3-Cat; Well 3, LysM3-chitin-hydrolytic inactive mutant; Well 4, LysM3; Well 5, a mixture of LysM3 and Cat; Well 6, a mixture of LysM3 and chitin-hydrolytic inactive mutant. Micrographs of the mycelium of *Trichoderma viride* before (**B**,**D**) and after (**C**,**E**) the addition of each protein solution; LysM3 (**B**,**C**) or LysM3-Cat (**D**,**E**). The black and white arrows indicate the disruption sites at the hyphal tips and those at the lateral walls around the septum, respectively. Scale bars = 100 μm. (Reprinted with permission from [78] Copyright © 2020, Elsevier.)

**Figure 2 biology-12-00087-f002:**
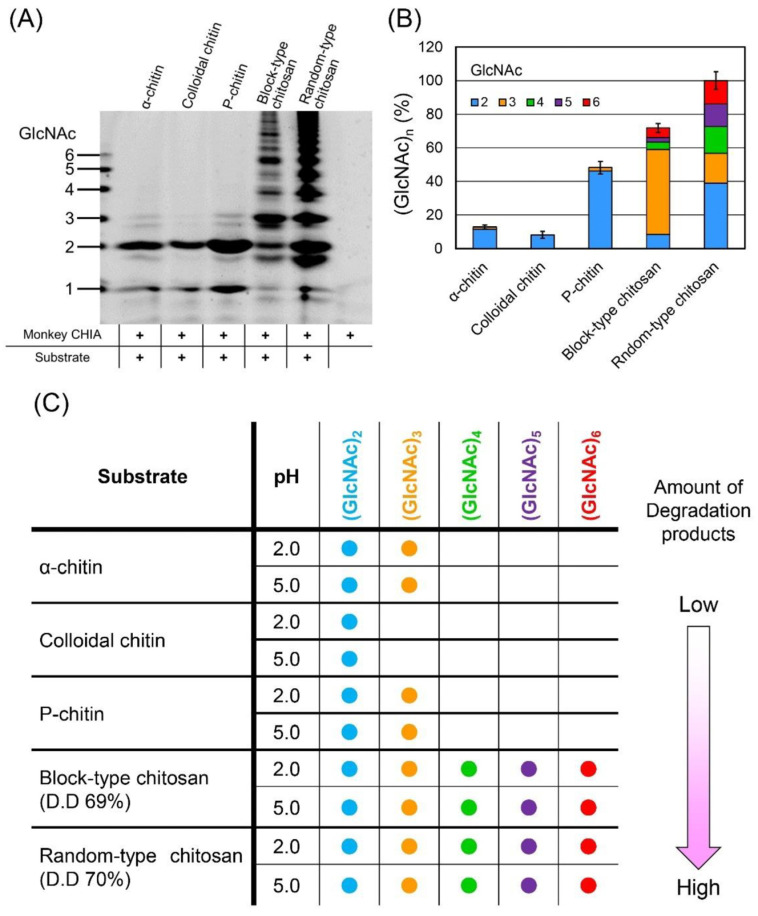
Comparison of degradation patterns of chitin and chitosan substrates. Five substrates are incubated with monkey *ChiA* at pH 5.0 and 50 °C for 72 h. Those include α-chitin, colloidal chitin, polymeric form chitin (P-chitin), and block-type and random-type chitosan. (**A**) Fluorophore-assisted carbohydrate electrophoresis analysis of chitin and chitosan degradation by monkey *ChiA*. Size standards on the left margin are defined as chitin oligomers. (**B**) Quantitative data of (GlcNAc)_2–5_ are obtained from each substrate. The quantitative data that express the percentage of the maximum signal of degradation products (the total amount of degradation products from random-type chitosan) are set to 100%. (**C**) Qualitative analysis of the produced COSs from each substrate. (Reprinted with permission from [90] Copyright © 2022 by Uehara and Takasaki et al. Licensee MDPI, Basel, Switzerland.)

**Figure 3 biology-12-00087-f003:**
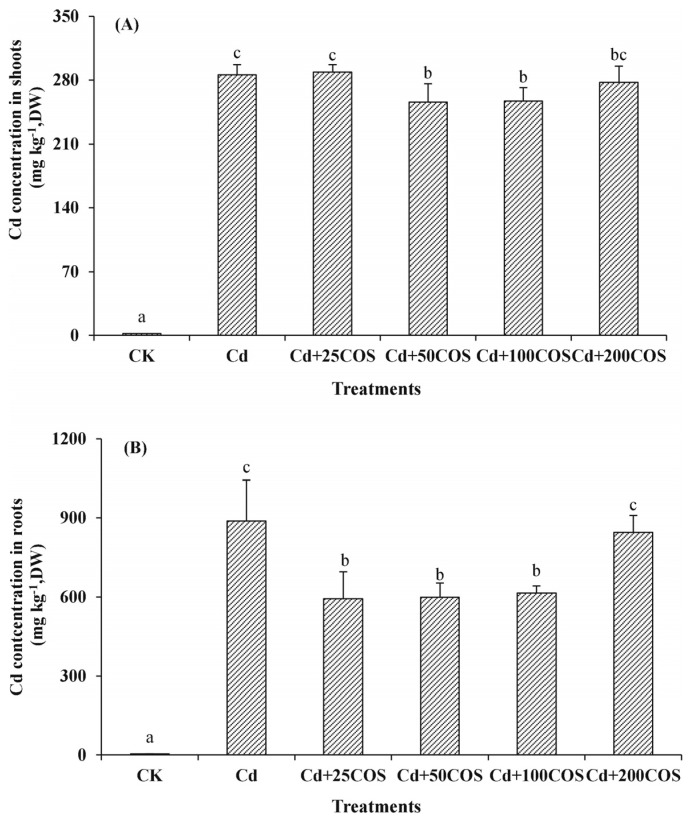
Effects of COS on cadmium (Cd) concentration in shoots (**A**) and roots (**B**) of edible rape (*Brassica rapa* L.) under Cd stress. After Cd treatment for 7 days, plants are sprayed with different concentrations of COS treatments: (1) control (CK), foliar spray of distilled water; (2) Cd, 50 μM Cd + foliar spray of distilled water; (3) Cd+ 25COS, 50 μM Cd + foliar spray of 25 mg/L COS; (4) Cd+ 50COS, 50 μM Cd + foliar spray of 50 mg/L COS; (5) Cd+ 100COS, 50 μM Cd + foliar spray of 100 mg/L COS; and (6) Cd + 200COS, 50 μM Cd + foliar spray of 200 mg/L COS. Different letters indicate significant differences at *p* < 0.05. (Reprinted with permission from [145] Copyright © 2017, Elsevier.)

**Figure 4 biology-12-00087-f004:**
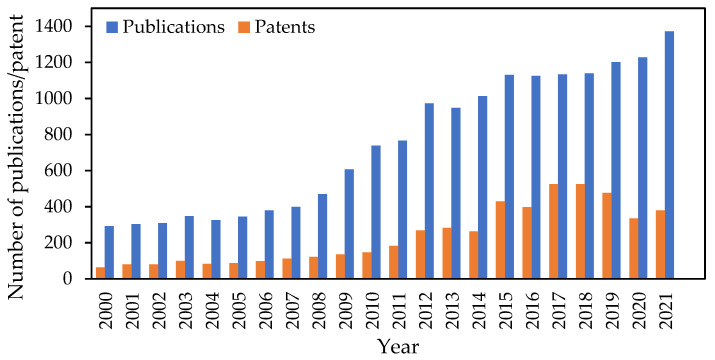
The number of publications and patents each year from 2000 to 2022, found in the SciFinder^TM^ database containing the concepts of “chitinase”, “chito-oligosaccharide”, and “chitinase and chito-oligosaccharide”.

**Figure 5 biology-12-00087-f005:**
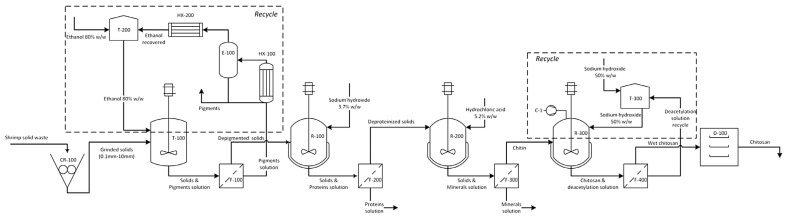
Process diagram for chitin and chitosan production. (Reprinted with permission from [156] Copyright © 2017, Elsevier.)

**Table 1 biology-12-00087-t001:** Bacterial source of chitinase for the production of GlcNAc and COSs ((GlcNAc)_n_) from different chitin substrates.

Chitinase Source	Substrate	Yield of (GlcNAc)_n_ (mg/g) ^1^	References
*Bacillus subtilis*	α-chitin, β-chitin, crude crab shell powder, chitosan	163 mg GlcNAc/g	[46,48]
*Bacillus atrophaeus BSS*	Colloidal chitosan	806 mg (GlcNAc)_2–6_/g	[47]
*Chitiniphilus shinanonensis*	Shrimp/ squid pen flakes	10.6 mg (GlcNAc)/g 62 mg (GlcNAc)/g	[49]
*Chitinolyticbacter meiyuanensis SYBC-H1*	Shrimp chitin powder	982 mg (GlcNAc)/g	[50]
*Enterobacter cloacae* subsp. *cloacae*	Colloidal chitin	0.405 mg (GlcNAc)/g 1.06 mg (GlcNAc)_2_/g	[51]
*Exiguobacterium antarcticum*	Crayfish shell chitin	761 mg (GlcNAc)_1–2_/g	[52]
*Flavobacterium johnsoniae UW101*	Colloidal chitin	59 mg (GlcNAc)_2_/g 47 mg (GlcNAc)_3_/g	[53]
*Paenibacillus* sp *LS 1*	Colloidal chitin (α, β)	53 mg (GlcNAc)_1–2_/g 721 mg (GlcNAc)_1–2_/g	[54]
*Paenicibacillus barengoltzii*	Crab shell, colloidal	720 mg (GlcNAc)_2_/g	[55,56]
*Streptomyces albolongus*	Colloidal chitin	2.8 mg (GlcNAc)/g	[57]
*Streptomyces chilikensis RC1830*	Colloidal chitin	761 mg (GlcNAc)_1–2_/g	[58,59]
*Streptomyces diastaticus CS1801*	Colloidal chitin	18.5 mg (GlcNAc)_1–5_/g	[60]
*Streptomyces sampsonii XY 2–7*	Shrimp powder	720 mg (GlcNAc)_1–2_/g	[61]
*Salinivibrio BAO-1801*	Shrimp shell	105 mg (GlcNAc)/g 715 mg (GlcNAc)_2_/g	[62]
*Thermomyces lanuginosus*	Shrimp shell	80 mg (GlcNAc)/g 720 mg (GlcNAc)_2_/g	[63]
*Vibrio campbellii* *(formerly V. harveyi)*	Shrimp flakes	200 mg (GlcNAc)_2_/g	[64]

^1^ Report as yield by mass of GlcNAc and COS product/mass of chitin substrate.

## Data Availability

Not applicable.

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
