# Peer review of "Chitinase-Assisted Bioconversion of Chitinous Waste for Development of Value-Added Chito-Oligosaccharides Products"

_biology, 2023, doi:10.3390/biology12010087_

Round 1

Reviewer 1 Report

This is a timely review of the use of chitinase from various sources for the bioconversion of chitinous waste. The reviewe is well written, with a complete references list acknowledging the recent studies in the research field. I enjoyed reading this review and I believe it will inspire many other readers as well. Therefore, I recommend publication after addressing a minor critique that will further improve the structural description of chitin.

Line 46-48, the alpha, beta, and gamma forms of chitin macromolecules are summarized. However two structural aspects need to be mentioned briefly here.

First, there are debates whether the gamma form is anew form or is an analogue of the alpha form but with mixed packing. This concept has been mentioned in an earlier review:

Rinaudo, M. Chitin and Chitosan: Properties and Applications. Prog. Polym. Sci. 31. 603-632 (2006). DOI: j.progpolymsci.2006.06.001

Second, the alpha, beta, and gamma forms are mainly determined using the high-crystallinity and purified chitin materials. Recent studies have revealed that the chitin structure in native environments and cell walls are much more complicated than the simplified models, which might be caused by the complexity of chitin synthesis (and the many families of chitin synthases involved) and likely due to the complex environments of chitin when it is placed together with many other biomolecules to form the cell wall. This concept has been revealed quite recently in multiple high-resolution structural studies:

Chakraborty et al. Nat. Commun. 12, 6346. DOI: 10.1038/s41467-021-26749-z

Ghassemi et al. Chem. Rev. 122, 10036-10086 (2022). DOI: 10.1021/acs.chemrev.1c00669

In general, this is a well-written review, with no apparent weakness. I believe it will benefit the research programs of many researchers in the field of chitin/chitosan, enzyme, and bioconversion.

Author Response

Response to Reviewers’ Comments

Manuscript ID: biology-2093307
Type of manuscript: Review
Title: Chitinase-assisted bioconversion of chitinous waste for development of value-added Chito-oligosaccharides products

Our detailed responses (in plain blue text) to the reviewers’ comments (in black italic) are provided below.

We thank the reviewers for the reviews and providing us insightful comments. We have made all the changes suggested by the reviewers in the revised manuscript. Revisions to the manuscript are marked up using the “Track Changes” function. The detailed response to each of the reviewers’ comments is given below, point-by-point.

Reviewer #1  

This is a timely review of the use of chitinase from various sources for the bioconversion of chitinous waste. The reviewe is well written, with a complete references list acknowledging the recent studies in the research field. I enjoyed reading this review and I believe it will inspire many other readers as well. Therefore, I recommend publication after addressing a minor critique that will further improve the structural description of chitin.

We thank the reviewer for the careful review to improve the manuscript.

Line 46-48, the alpha, beta, and gamma forms of chitin macromolecules are summarized. However two structural aspects need to be mentioned briefly here.

First, there are debates whether the gamma form is anew form or is an analogue of the alpha form but with mixed packing. This concept has been mentioned in an earlier review:

Rinaudo, M. Chitin and Chitosan: Properties and Applications. Prog. Polym. Sci. 31. 603-632 (2006). DOI: j.progpolymsci.2006.06.001

We thank the reviewer for pointing this out. We have mentioned structural aspects and the recent study about γ-chitin. However, the structure of γ-chitin cannot be delineated. Therefore, the statement whether the γ form is a new form or is an analogue of the α form has also been mentioned in the manuscript.

Please see lines 48-53 in the revised manuscript. 

Second, the alpha, beta, and gamma forms are mainly determined using the high-crystallinity and purified chitin materials. Recent studies have revealed that the chitin structure in native environments and cell walls are much more complicated than the simplified models, which might be caused by the complexity of chitin synthesis (and the many families of chitin synthases involved) and likely due to the complex environments of chitin when it is placed together with many other biomolecules to form the cell wall. This concept has been revealed quite recently in multiple high-resolution structural studies:

Chakraborty et al. Nat. Commun. 12, 6346. DOI: 10.1038/s41467-021-26749-z

Ghassemi et al. Chem. Rev. 122, 10036-10086 (2022). DOI: 10.1021/acs.chemrev.1c00669

We have included these two references to discuss more about chitin forms to improve the Introduction. This will provide interesting continuous research that may be of interest to readers.

Please see lines 54-57 in the revised manuscript.

In general, this is a well-written review, with no apparent weakness. I believe it will benefit the research programs of many researchers in the field of chitin/chitosan, enzyme, and bioconversion.

We thank the reviewer for all the suggestions. We hope that we understand the reviewer’s questions correctly.

Reviewer 2 Report

In this manuscript, the authors describe an interesting topic “Chitinase-assisted bioconversion of chitinous waste for development of value-added Chito-oligosaccharides products”. However, there are many misinformations provided throughout the manuscript and they should be revised (see below) to move forward.

Line 17: the authors mentioned that they discussed the patents published to date. I didn’t see many patents in the manuscript. If it is true, the authors should provide a table with the patents, including patent No. Or, if there are not too many patents, they could simply delete this word.  

Lines 19 and 43: Chitin found in the cell walls of fungi – this is not true. It seems that the authors didn’t search well on current literature. The exoskeletons of invertebrates – no ref provided. Please have a look at public literature to find more details about cell walls and chitin that very clearly demonstrate the chitin in coralline algae, but not only fungi, e.g., https://www.nature.com/articles/srep06162

Line 46: Ref 9 is not 100% relevant for solubilizing agents.

Lines 76-78: “This strategy provides COSs as the products that are environmentally friendly and safe for human uses, enabling to be applied in various fields of agriculture, biotechnology and biomedicine” Ref is needed, especially for agriculture applications.

Lines 82-84: “The recent research literatures on new chitinase from various sources for COS production, COS-derivative synthesis, updated applications, and commercialization of COSs are summarized in this review”. The content here is not true. The authors should find more relevant reports and remove irrelevant references to minimize the length of the mauscript.  

Also, this manuscript is too long. The authors could switch some less important data to suppl. info.

Fig. 2A: MW weight must be labeled for each band.  

English errors throughout the manuscript should be fixed. 

Author Response

Response to Reviewers’ Comments

Manuscript ID: biology-2093307
Type of manuscript: Review
Title: Chitinase-assisted bioconversion of chitinous waste for development of value-added Chito-oligosaccharides products

Our detailed responses (in plain blue text) to the reviewers’ comments (in black italic) are provided below.

We thank the reviewers for the reviews and providing us insightful comments. We have made all the changes suggested by the reviewers in the revised manuscript. Revisions to the manuscript are marked up using the “Track Changes” function. The detailed response to each of the reviewers’ comments is given below, point-by-point.

Reviewer #2

In this manuscript, the authors describe an interesting topic “Chitinase-assisted bioconversion of chitinous waste for development of value-added Chito-oligosaccharides products”. However, there are many misinformations provided throughout the manuscript and they should be revised (see below) to move forward.

We thank the reviewer for the careful review to improve the manuscript.

Line 17: the authors mentioned that they discussed the patents published to date. I didn’t see many patents in the manuscript. If it is true, the authors should provide a table with the patents, including patent No. Or, if there are not too many patents, they could simply delete this word. 

Since there are not many patents as compared to the academic publications in this review. We simply deleted the word according to the reviewer’s suggestion.

Lines 19 and 43: Chitin found in the cell walls of fungi – this is not true. It seems that the authors didn’t search well on current literature.

The chitin content of the fungal wall varies according to the morphological phase of the fungus. It represents 1–2% of the dry weight of yeast cell wall while in filamentous fungi, it can reach up to 10–20%[1]. In fungi, chitin serves as the structural polysaccharide stabilizing the cell wall [2] and can preserve the fungal cells even at high temperature [3]. The chitin found to be the most polymorphic molecule in fungal cell walls is also studied by solid-state NMR spectroscopy, which reveals the presence of chitin in extracellular matrixes (ECMs), such as the cell walls [4] the distribution of chitin and α-1,3-glucan build a hydrophobic scaffold, contributing to cell wall rigidity [5].

Chitin is found in three locations: a ring at mother-bud neck, the primary septum and throughout the lateral cell wall [6]. Chitin-directed organization of the cell wall layers allows the fungal cell to effectively monitor and interact with the external environment [7]. Signaling pathways identified in fungi that contribute to the maintenance of the cell wall during fungal growth and activation of chitin synthesis in response to changes in the external environment provoking cell wall stress can be found in Fig. 4 of this review reported by Merzendorfer, 2011 [8]. However, chitin is not found in all fungi and may be absent in one species that is closely related to another [9].

References

  1. R. Garcia-Rubio, H. C. de Oliveira, J. Rivera, and N. Trevijano-Contador, The Fungal Cell Wall: Candida, Cryptococcus, and Aspergillus Species. Frontiers in Microbiology, 2020. Vol. 10.
  2. Jean-Paul Latgé, The cell wall: a carbohydrate armour for the fungal cell. Molecular Microbiology, 2007. Vol. 66 pp. 279-290.
  3. Shigeru Deguchi, Kaoru Tsujii, and Koki Horikoshi, In situ microscopic observation of chitin and fungal cells with chitinous cell walls in hydrothermal conditions. Scientific Reports, 2015. Vol. 5 pp. 11907.
  4. Nader Ghassemi, Alexandre Poulhazan, Fabien Deligey, Frederic Mentink-Vigier, Isabelle Marcotte, and Tuo Wang, Solid-State NMR Investigations of Extracellular Matrixes and Cell Walls of Algae, Bacteria, Fungi, and Plants. Chemical Reviews, 2022. Vol. 122 pp. 10036-10086.
  5. Xue Kang, Alex Kirui, Artur Muszyński, Malitha C. Dickwella Widanage, Adrian Chen, Parastoo Azadi, Ping Wang, Frederic Mentink-Vigier, and Tuo Wang, Molecular architecture of fungal cell walls revealed by solid-state NMR. Nature Communications, 2018. Vol. 9 pp. 2747.
  6. Javier Arroyo, Vladimír Farkaš, Ana Belén Sanz, and Enrico Cabib, ‘Strengthening the fungal cell wall through chitin–glucan cross-links: effects on morphogenesis and cell integrity’. Cellular Microbiology, 2016. Vol. 18 pp. 1239-1250.
  7. H. E. Brown, S. K. Esher, and J. A. Alspaugh, Chitin: A “hidden figure” in the fungal cell wall, in Current Topics in Microbiology and Immunology. 2020. p. 83-111.
  8. Hans Merzendorfer, The cellular basis of chitin synthesis in fungi and insects: Common principles and differences. European Journal of Cell Biology, 2011. Vol. 90 pp. 759-769.
  9. Mostafa M. Abo Elsoud and E. M. El Kady, Current trends in fungal biosynthesis of chitin and chitosan. Bulletin of the National Research Centre, 2019. Vol. 43 pp. 59.

The exoskeletons of invertebrates – no ref provided.

We have cited more references to support the statement.

Please see Line 44 in the revised manuscript.

Please have a look at public literature to find more details about cell walls and chitin that very clearly demonstrate the chitin in coralline algae, but not only fungi, e.g., https://www.nature.com/articles/srep06162

We thank the reviewer for this info. we have included this reference (ref. 13) in Line 43.

Line 46: Ref 9 is not 100% relevant for solubilizing agents.

We have replaced this reference with other relevant references.

Please see Line 46 in the revised manuscript.

Lines 76-78: “This strategy provides COSs as the products that are environmentally friendly and safe for human uses, enabling to be applied in various fields of agriculture, biotechnology and biomedicine” Ref is needed, especially for agriculture applications.

We thank the reviewer for pointing this out. We have cited the references.

Please see Line 87-88 in the revised manuscript.

Lines 82-84: “The recent research literatures on new chitinase from various sources for COS production, COS-derivative synthesis, updated applications, and commercialization of COSs are summarized in this review”. The content here is not true. The authors should find more relevant reports and remove irrelevant references to minimize the length of the mauscript.

 Also, this manuscript is too long. The authors could switch some less important data to suppl. info.

This review provided an overview of the use of chitinase from various sources recently reported, with a special focus on COS production, which is different from other reviews, articles or book chapters related to the role of chitinase in COS production. We also further provided an update application of COS, which mainly have been used as antimicrobial agents similar to chitosan. We have replaced some references with more relevant ones.

We think that the length of the manuscript is appropriate for the full-length review.

Fig. 2A: MW weight must be labeled for each band. 

We really apologize regarding that MW cannot be labeled. This figure is reprinted from ref. 90.

English errors throughout the manuscript should be fixed.

The manuscript has been proofread by native English speaker according to the reviewer suggestion.

We thank the reviewer for all the suggestions. We hope that we understand the reviewer’s questions correctly.

Round 2

Reviewer 2 Report

N/A